# Ancient DNA Methods Improve Forensic DNA Profiling of Korean War and World War II Unknowns

**DOI:** 10.3390/genes13010129

**Published:** 2022-01-11

**Authors:** Elena I. Zavala, Jacqueline Tyler Thomas, Kimberly Sturk-Andreaggi, Jennifer Daniels-Higginbotham, Kerriann K. Meyers, Suzanne Barrit-Ross, Ayinuer Aximu-Petri, Julia Richter, Birgit Nickel, Gregory E. Berg, Timothy P. McMahon, Matthias Meyer, Charla Marshall

**Affiliations:** 1Department of Evolutionary Genetics, Max Planck Institute for Evolutionary Anthropology, 04103 Leipzig, Germany; aximu_ayinuer@eva.mpg.de (A.A.-P.); julia_richter@eva.mpg.de (J.R.); nickel@eva.mpg.de (B.N.); mmeyer@eva.mpg.de (M.M.); 2Armed Forces Medical Examiner System’s Armed Forces DNA Identification Laboratory (AFMES-AFDIL), Dover Air Force Base, Dover, DE 19902, USA; jacqueline.t.thomas2.ctr@mail.mil (J.T.T.); kimberly.s.andreaggi.ctr@mail.mil (K.S.-A.); jennifer.l.higginbotham3.ctr@mail.mil (J.D.-H.); kerriann.k.meyers.ctr@mail.mil (K.K.M.); suzanne.m.barritt-ross.civ@mail.mil (S.B.-R.); Timothy.p.mcmahon10.civ@mail.mil (T.P.M.); 3SNA International, Contractor Supporting the Armed Forces Medical Examiner System, Alexandria, VA 22314, USA; 4Department of Immunology, Genetics and Pathology, Uppsala University, SE-751 08 Uppsala, Sweden; 5Defense Personnel Accounting Agency, Central Identification Laboratory, Hickam Air Force Base, Oahu, HI 96853, USA; gregory.e.berg2.civ@mail.mil; 6Forensic Science Program, Pennsylvania State University, State College, PA 16802, USA

**Keywords:** degraded DNA, massively parallel sequencing (MPS), mitochondrial DNA, forensic DNA profiling, ancient DNA, human identification

## Abstract

The integration of massively parallel sequencing (MPS) technology into forensic casework has been of particular benefit to the identification of unknown military service members. However, highly degraded or chemically treated skeletal remains often fail to provide usable DNA profiles, even with sensitive mitochondrial (mt) DNA capture and MPS methods. In parallel, the ancient DNA field has developed workflows specifically for degraded DNA, resulting in the successful recovery of nuclear DNA and mtDNA from skeletal remains as well as sediment over 100,000 years old. In this study we use a set of disinterred skeletal remains from the Korean War and World War II to test if ancient DNA extraction and library preparation methods improve forensic DNA profiling. We identified an ancient DNA extraction protocol that resulted in the recovery of significantly more human mtDNA fragments than protocols previously used in casework. In addition, utilizing single-stranded rather than double-stranded library preparation resulted in increased attainment of reportable mtDNA profiles. This study emphasizes that the combination of ancient DNA extraction and library preparation methods evaluated here increases the success rate of DNA profiling, and likelihood of identifying historical remains.

## 1. Introduction

Massively parallel sequencing (MPS), also known as next-generation sequencing (NGS), has only recently begun integration into the forensic field. MPS was first accepted in a court case in the Netherlands in 2019 [1]. The technology has also been approved for use within the U.S. criminal justice system for both STR profiling and mitochondrial (mt) DNA sequencing [2]. Furthermore, whole genome (shotgun) sequencing and genome-wide SNP arrays are often used for investigative genetic genealogy to provide leads in active and cold cases [3]. While more challenging cases involving historical skeletal remains, including historical figures, unmarked graves, and unidentified war victims, have seen some advances in DNA profiling with MPS, success rates are still low [4,5,6]. This may be due to the fact that successful forensic DNA profiling typically requires relatively large DNA fragments (greater than ~100 base pairs) with current methods, which are not always available in sufficient number in degraded samples.

One set of historical remains that exemplifies the challenges faced by historical identification cases is Unknowns from the Korean War buried in the National Memorial Cemetery of the Pacific (NMCP) in Honolulu, Hawaii [7]. These remains contain poorly preserved DNA due to extensive chemical treatment that was standard postmortem military practice in the 1940s and 1950s. The war dead were initially collected in temporary graves (Figure 1a) by the Army Graves Registration Service. They were then transferred to the United Nations Cemetery in Tanggok, Republic of Korea [8,9] where they were treated with insecticide and deodorant. The remains were subsequently shipped to the Army’s Central Identification Unit in Kokura, Japan [7,10]. At this location, the remains were soaked in 40–50% formaldehyde solution for three to five days (Figure 1b). Formaldehyde is known to lead to the formation of cross-linked complexes between DNA molecules or between DNA and proteins, complicating DNA extraction [11]. In addition, Hawaii has a warm and wet environment, while Korea is climatically diverse with temperatures fluctuating from freezing to extremely hot. Such environmental conditions involving high heat have been found to increase DNA degradation, specifically deamination and fragmentation rates [12]. After further treatment with fungicide and a powdered hardening compound (Figure 1c), bodies were wrapped and casketed. In total, the remains of 859 unidentified Korean War service members were shipped to Hawaii in 1956 and buried as Unknowns at the NMCP, also known as the Punchbowl [7]. In 1999, the then Central Identification Laboratory, Hawaii (CILHI, now called the Defense POW/MIA Accounting Agency (DPAA)) [13], undertook a fledgling exhumation program focused on identifying Korean War Unknowns at NMCP. Working in concert with the CILHI, the Armed Forces DNA Identification Laboratory (a division of the Armed Forces Medical Examiner System (AFMES-AFDIL)) attempted DNA testing of the first five exhumed cases using standard forensic DNA profiling techniques [14]. However, the results were not reproducible and therefore determined to be unreliable.

In 2016 the AFMES-AFDIL validated a novel MPS protocol for mitochondrial genome (mitogenome) sequencing using a probe capture and Illumina sequencing approach. This method was shown to be more amenable to the crosslinked and severely fragmented DNA from NMCP Unknowns, which was detailed in a 2017 performance evaluation [4]. As a result of this MPS mitogenome sequencing validation, the AFMES-AFDIL became the first accredited laboratory to adopt MPS for forensic casework worldwide. The study showed severe degradation of the DNA from Korean War NMCP Unknowns, which averaged <70 base pairs (bp) in length. The validated MPS methods are now routinely applied to all chemically treated or severely degraded DNA samples at the AFMES-AFDIL, regardless of the military conflict. These include remains from U.S. military WWII cemetery exhumations with similar preservation problems to those encountered with the Korean War unknowns from the NMCP. To date, over 3000 samples have been tested using the validated mitogenome MPS protocol with a success rate of ~60%. While this is a marked improvement from before the integration of MPS into the casework workflow (6% success rate with PCR and Sanger sequencing), the sample failure rate is still substantial, even when targeting mtDNA, which is high in copy number relative to nuclear DNA. Additional advancements to improve MPS sensitivity could elevate success rates in reportable mitogenome profile generation and may furthermore enable SNP profiling for extended kinship analysis [15].

While the forensic field has faced obstacles in generating DNA profiles from historical remains, the ancient DNA field has been successful in recovering authentic DNA from very old and degraded samples from a variety of substrates and environments. Advancements in the extraction [16,17], library preparation [18], and enrichment of target DNA [19,20] have enabled the generation of hominin nuclear and mtDNA sequence profiles from skeletal elements as well as sediment over 300,000 years old [21,22,23]. Of particular importance in these developments was an extraction protocol that can recover DNA fragments <50 bp in length [16] and transitioning from double- to single-stranded DNA library preparation [24], a method that has also been shown to improve the recovery of DNA molecules from formalin-fixed samples [25,26,27]. However, the focus for method development for forensic casework and ancient DNA studies is different. This is due to the consistent, degraded nature of ancient DNA samples that motivates the foundation for methodological advancements in the field. Forensic samples, by contrast, are much more heterogeneous in terms of DNA quality. Novel forensic methods are largely aimed to improve DNA profiling of traditional forensic targets that exceed 100 bp in size. Furthermore, new forensic DNA methods must meet rigorous validation standards, making it difficult to test and adopt new protocols. These differing forces that drive the two fields have positioned ancient DNA considerably ahead of forensics in terms of method development. Yet both fields share the common problem of analyzing DNA that is (or can be) highly degraded. This commonality prompted the question if methods from the ancient DNA field could improve success rates for generating mtDNA profiles for historical identifications, which has been suggested in recent studies and reviews [28,29,30]. Therefore, we tested the feasibility of integrating ancient DNA methods into a forensic casework workflow by comparing different DNA extraction and library preparation protocols using fifteen skeletal remains with DNA of varying quality as determined by previous testing. We then evaluated the success of each protocol based on the number of informative sequences, completeness of mtDNA profiles, and concordance with expected profiles when available. This study lays the groundwork for further developments that will advance DNA profiling and identification of the poorest quality historical remains samples.

## 2. Materials and Methods

### 2.1. Skeletal Samples

Fifteen non-probative skeletal samples were utilized for DNA testing in this study (Table 1). The samples originated from disinterred burials of World War II and Korean War service members that were likely to have been chemically treated, either with formalin or with chemical powder (deodorant or insecticide) application. All fifteen samples were previously processed in routine casework at the AFMES-AFDIL using the validated mitogenome sequencing methods outlined in [4]. Samples were selected based on DNA quality in order to cover a range in quality types and not based on skeletal element. The samples ranged in DNA quality based on casework sequencing success: seven failed with <10-fold average coverage, four provided mtDNA sequences with 10-35-fold (low) average coverage, and four yielded high coverage mtDNA data (100-360-fold).

### 2.2. DNA Extraction

All DNA extractions were performed at the AFMES-AFDIL in a clean, pre-PCR room with positive pressure, per standard ancient and forensic DNA practices. Under sterile conditions, each bone was sanded to remove the external surface, washed in ethanol, and then powdered using a blender cup. Bone powder samples were digested and a reagent blank (RB) was initiated for each extraction set. DNA was extracted using one of the conditions outlined in Table 2. As described in Edson et al. [32], the AFDIL method utilized 1.0 g of bone powder incubated overnight at 56 °C in 7.5 mL digestion buffer (0.5 M EDTA, 1% lauroylsarcosine) with 200 µL proteinase K (20 mg/mL). DNA was then isolated using an organic DNA extraction protocol with phenol chloroform isoamyl alcohol (PCIA), followed by concentration and buffer exchange using an Amicon Ultra-4 (molecular weight cut-off 30 kDa) (MilliporeSigma, Burlington, MA, USA). The resulting DNA extracts were divided into three aliquots to assess the effect of DNA repair on the sequence data. The following repair conditions were evaluated: (1) no repair, (2) USER (New England Biolabs (NEB), Ipswich, MA, USA) treatment to remove uracil residues resulting from cytosine deamination, and (3) NEBNext FFPE DNA Repair Mix (NEB) to remove uracils and also repair nicks, gaps, oxidized bases, and blocked 3′ ends [33]. Repair treatments followed their respective manufacturer protocols. All three DNA extracts (no repair, USER, and FFPE DNA Repair) underwent MinElute (QIAGEN, Hilden, Germany) purification after repair. This resulted in a total of three different PCIA-based protocols that were tested (PM: no repair, PUM: USER, and PFM: FFPE DNA Repair).

The ancient DNA extraction protocol followed was the Dabney method [16] as described in Rohland et al. [17], substituting the binding buffer ‘D’ with QIAGEN PB buffer. This method utilized 0.2 g bone powder digested at 37 °C in 1 mL 0.45 M EDTA with 0.05% Tween 20 and 25 µL proteinase K (20 mg/mL). Unlike the AFDIL digestion, there was not a complete digestion of bone powder. Therefore, the bone powder pellets remaining after the initial 37 °C Dabney extraction were re-digested overnight with 1.0 mL buffer at 56 °C, which resulted in the powder going into solution. Purification was then performed with the Roche High Pure Spin Column (a large volume silica column) (Pleasanton, CA, USA). This resulted in two ancient DNA extractions, although each were from the same bone powder aliquot (first digestion: aDNA37 and second digestion: aDNA56).

Finally, a hybrid “AFDIL-Dabney” DNA extraction method was performed following the AFMES-AFDIL digestion procedures (i.e., 1.0 g of bone powder incubated overnight at 56 °C in 7.5 mL digestion buffer and 200 µL proteinase K) and Dabney purification (i.e., High Pure Spin Column).

### 2.3. Library Preparation

#### 2.3.1. In House MPI Method

This single-stranded DNA library preparation was performed as described in [19] on a Bravo NGS Workstation (Agilent Technologies) at the Max Planck Institute for Evolutionary Anthropology (MPI-EVA) in Leipzig, Germany from 5µL of each sample extract and RB. In short, the DNA was denatured by heat, and then the 5′ and 3′ phosphate groups were removed. T4 DNA ligase was used to ligate an adapter oligonucleotide with a 3′ biotin to the 3′ end of each DNA fragment, aided by a splinter oligonucleotide. This product was then immobilized on streptavidin magnetic beads and a wash was performed at 45 °C to remove the splinter. Primer and the *E.coli* DNA polymerase I Klenow fragment was added to make a copy of the template strand from the attached adapter. A bead wash was then performed to remove excess primers and prevent adapter dimer formation. Next, T4 DNA ligase was used for a blunt-end ligation of a second, double-stranded adapter. Heat denaturation at 95 °C was then used to release the library products from the beads. Library amplification and indexing was performed with AccuPrime Pfx DNA polymerase with the following cycling parameters: 95 °C for 2 min, then 35 cycles of 95 °C for 20 s, 60 °C for 30 s and 68 °C for 1 min, with a final additional 5 min at 68 °C. Post-amplification purification was performed using the MinElute PCR purification kit. Libraries were generated from all 15 samples, one RB, one library positive control and one library negative control.

#### 2.3.2. SRSLY Kit

The SRSLY single-stranded DNA library preparations were performed manually at the AFMES-AFDIL using Claret’s SRSLY PicoPlus Library Prep Kit for Illumina (Santa Cruz, CA, USA). Unless otherwise noted, library preparation was carried out according to the manufacturer protocol for ancient and highly degraded DNA samples, including purification steps with AMPure XP (Beckman Coulter, Brea, CA, USA) for maximum short fragment retention [34]. In summary, 5 µL of extract was used for library preparation, and a positive control and negative control were included with each sample set. The positive control was enzymatically fragmented K562 cell line with 1 ng library input, as described in [4]. An initial denaturation step rendered all DNA single-stranded and then stabilized by a single-stranded binding protein, followed by adapter ligation at 37 °C (lid 45 °C) for 1 h with splinted adapters that retain the original strand orientation. AMPure XP purification after adapter ligation included the addition of isopropanol and 10 mM Tris-HCl pH 8.5 with a final ratio of 0.54× (0.99× ratio to library prior to isopropanol and Tris-HCl addition). The full library volume was amplified in a single reaction using Claret unique dual indexes (undiluted) and KAPA HiFi HotStart Uracil+ ReadyMix (Roche) according to the manufacturer recommendations. Uracil+ was selected in order to tolerate uracil bases present in the DNA due to cytosine deamination. Library amplification was completed with the following parameters: 95 °C for 3 min, then 35 cycles of 95 °C for 20 s, 60 °C for 15 s, 72 °C for 1 min, followed by 72 °C for 7 min. Post-amplification purification utilized a 1.2× AMPure XP ratio. Libraries were generated from a total of 15 samples, one RB, two positive controls and two negative controls.

#### 2.3.3. KAPA Hyper Prep Kit

The KAPA double-stranded DNA library preparations were performed manually at the AFMES-AFDIL using the Roche KAPA Hyper Prep Kit. Unless otherwise noted, library preparation was carried out according to the manufacturer protocol. Again, 5 µL of extract was used for library preparation, and a positive control and negative control were included with each sample set. The positive control was fragmented DNA from a K562 cell line with 1 ng library input. End repair and A-tailing took place in a single step with incubation at 20 °C for 30 min, then 65 °C for 30 min (lid 85 °C), followed by adapter ligation using 15 µM KAPA dual indexed adapters for all libraries at 20 °C for 15 min. Post-ligation clean-up used an increased 2× AMPure XP ratio in order to retain smaller fragments. The full library volume was amplified using KAPA HiFi HotStart Uracil+ ReadyMix and KAPA Library Amplification Primer mix, according to the manufacturer’s recommendations with the following parameters: 95 °C for 3 min, then 35 cycles of 98 °C for 20 s, 60 °C for 15 s, 72 °C for 1 min, followed by 72 °C for 7 min. The same post-amplification AMPure XP purification was utilized as with the SRSLY libraries. Libraries were generated from a total of 15 samples, one RB, two positive controls and two negative controls.

#### 2.3.4. Quantification

To compare the number of DNA molecules converted into library molecules for each of the three library preparation methods, quantitative (q) PCR was performed on the unamplified library products of each method. The qPCR was performed as described in Gansauge et al. [18] at the MPI-EVA with all the libraries on the same plate to remove any possible plate-to-plate variation.

### 2.4. MtDNA Hybridization Capture

In-solution hybridization capture was used for the enrichment of human mtDNA for the three different types of library preparation methods. Two rounds of capture were performed following the protocol outlined in Fu et al. [19] in 384-well format on a Bravo NGS Workstation B (Agilent, Santa Clara, CA, USA) with adjustments to the number of amplification cycles per library to prevent each from reaching plateau.

### 2.5. Sequencing and Data Processing

For the comparison of extraction protocols, all resulting single-stranded DNA libraries were pooled for shotgun sequencing. For the comparison of library preparation protocols, each library type was pooled separately (three total pools) for shotgun sequencing and the human mtDNA enriched SRSLY and MPI library sets (two pools). In order to avoid captured libraries with identical indices being sequenced together, the KAPA Hyper Prep human mtDNA enriched libraries were split between nine pools and sequenced with other libraries not included in this study. All sequencing pools were created by combining equal amounts of sample libraries and 1:10 dilutions of controls (RBs and library negative controls). Library positive controls were not sequenced. Sequencing was performed on a MiSeq or NextSeq (Illumina technology) in 2× 75 bp paired-end configuration with 8-bp dual-indexed reads. Base calling was completed with bustard (Illumina) and overlapping paired-end reads were merged using leeHom [35]. For shotgun sequencing, reads were then mapped to the human reference genome GRCh37/1000 (Genomes release; ftp://ftp.1000genomes.ebi.ac.uk/vol1/ftp/technical/reference/phase2_reference_assembly_sequence/, accessed on 12 November 2021) and captured library reads were mapped to the revised Cambridge reference sequence [31]. PCR duplicates were then removed using bam-rmdup (https://github.com/mpieva/biohazard-tools, accessed on 15 October 2021).

For the processing of the shotgun results, all sequences less than 35 bp and unmapped sequences were removed. The remaining sequences were then used to calculate the average fragment size, rates of C-to-T substitutions, and number of informative sequences [26] in each sample. The 35 bp cut-off was used to allow comparisons to previous published ancient DNA data.

For the processing of the capture enriched data, the resulting de-duplicated sequences from each library preparation type continued either through the MPI workflow or a CLC Genomic Workbench (QIAGEN, Hilden, Germany) workflow. For the MPI workflow, unmapped sequences, sequences shorter than 30 bp or with a mapping quality less than 25 were removed. The resulting sequences were used to calculate the coverage of the mtDNA genome. For the CLC workflow, the de-duplicated bam files were ingested into the CLC at the AFMES-AFDIL for variant calling, mtDNA haplogroup prediction, and forensic profile reporting using AQME [36], in addition to determining the mtDNA coverage. A minimum read depth of 5 was required for bases to be included in the reported sequence range. Variant detection required a minimum of 4 reads supporting the variant as well as a minimum frequency of 10%. Point heteroplasmy was documented when variants exhibited 10–90% frequency. Length heteroplasmy was reported based on the major molecule; however, this was ignored in concordance assessments due to the impact of differing polymerases on length variants [37]. Additionally, variants consistent with cytosine deamination (C-to-T or G-to-A substitutions) that were low in frequency (<30%) and/or lacking in forward/reverse read balance (<0.30) were excluded. These variant detection parameters minimized the reporting of stochastic error while allowing for the mixture and contamination detection necessary in forensic casework. When processing the SRSLY libraries, contaminant pig DNA was identified as mismapping to the human mtDNA reference genome (rCRS), resulting in spikes of coverage and unexpected variants (Appendix A). The presence of pig sequences in these libraries was confirmed using a pipeline [38] that uses BLAST [39] and MEGAN [40] to assign sequences from sediment samples to different mammalian families (Appendix A). To remove these sequences a “pig-out” pipeline was added to both the CLC and MPI workflows as described in Appendix A.

## 3. Results

### 3.1. Evaluation of Extraction Methods

In order to evaluate the feasibility of integrating ancient DNA methods into a forensic casework workflow, we first evaluated the performance of various extraction protocols with respect to success in recovering DNA fragments. Fifteen non-probative skeletal remains that had been previously tested at the AFMES-AFDIL and were therefore of known quality (Table 1). We selected five different extraction/repair protocols to test: three PCIA-based from the forensic field [32], one from the ancient DNA field ([17], with PB buffer), and one that combined elements of each the forensic and ancient DNA protocols (see Section 2.2). As the bone powder pellet did not go into solution in the initial ancient DNA digestion at 37 °C, a redigestion was performed at 56 °C of the same pellet and each of the fractions was purified. Therefore, six extraction/repair conditions were tested in total (Table 2); however, the two ancient DNA extractions were from the same original powder sample.

DNA extraction performance was assessed using shotgun sequencing data produced from single-stranded libraries constructed following the method outlined in Gansauge et al. [18]. To determine which extraction protocol was most effective, we estimated the number of unique DNA fragments present in each library that mapped to the human DNA reference genome (informative sequences) in each library. For each sample we then determined which extraction protocol resulted in the most recovered informative sequences and took the ratio of the informative sequences from each other extraction protocol to this “best” protocol. This resulted in the relative informative sequences for each sample and extraction combination (see Appendix A for further explanation on how informative sequences and relative informative sequences were calculated). This calculation allowed us to compare the relative informative sequences across extraction types for the different samples (Figure 2A, see Appendix A for which extraction was best per sample). We found that the combined forensic ancient DNA extraction protocol (AFDIL-Dabney) was significantly worse than all other protocols (*n* = 15, Wilcoxon test with correction for multiple testing, *p*-values: 6.5 × 10^−^^4^ to 1.1 × 10^−^^4^). The redigestion at 56 °C with the ancient DNA protocol (aDNA56) recovered significantly more informative fragments than the two forensic extraction protocols that included extra DNA repair steps intended to improve recovery of damaged DNA (*n* = 15, Wilcoxon test with correction for multiple testing, *p*-values both 2.8 × 10^−^^2^). This loss of recovered DNA is likely, at least in part, due to the addition of extra purification steps. No significant difference in recovered informative sequences was observed between the two ancient DNA protocols and the forensic protocol with no DNA repair (*p*-values: 6.4 × 10^−^^1^ to 1). However, as the two ancient DNA protocols originate from the same bone powder pellet, theoretically if the initial lysis had been performed at 56 °C the DNA recovered in the 37 °C and 56 °C fractions would have been extracted together. If we combine the recovered informative sequences from both of these fractions, the ancient DNA protocol results in significantly more informative sequences than the forensic extraction protocols (*n* = 15, Wilcoxon test with correction for multiple testing, *p*-values all <4.8 × 10^−^^4^, Figure 2B). These results indicate that the ancient DNA extraction protocol does improve the recovery of target DNA from historical skeletal remains that are of interest for forensic casework particularly in a MPS workflow.

### 3.2. Characteristics of Recovered DNA

Single-stranded DNA library preparation coupled with shotgun sequencing allows for the characterization of nuclear DNA recovered from each of these samples. Specifically, we examined the percentage of sequences longer than 35 bp that mapped to the human genome, average fragment size for each sample, and deamination as represented by cytosine to thymine (C-to-T) substitutions. The latter two analyses were performed after removing unmapped reads, PCR duplicates, and sequences shorter than 35 bp or with mapping qualities less than 25. We used the data from the ancient DNA extraction protocol with a 56 °C incubation to investigate each of these factors.

The percent mapped for the samples ranged from 0.13–72.5% (mean 7.88%) and the average fragment size for each of the samples ranged from 44 to 64 bp (mean 49 bp). The observed percent of C-to-T substitutions on the 5′ ends ranged from 8.8% to 30.7% (mean 18.3%) and 8% to 22.3% (mean 13.3%) on the 3′ end. These values are similar to those observed in ancient DNA samples, where the presence of short DNA fragments and deamination are used for the authentication of ancient DNA [41] as well as to disentangle endogenous ancient DNA from modern human contamination [42]. While there have been multiple studies on the impacts of various factors on the DNA degradation patterns from ancient samples [12,22,43] and modern samples [44,45], there is limited data on DNA degradation in the more recent past. Therefore, we compared the average fragment size and deamination patterns of this set of forensic samples to published shotgun data from seven Late Pleistocene modern human specimens (~37 to 46,000 years ago) [46]. We found that the percent of sequences that mapped to the human reference genome (fragment length ≥ 35 bp and mapping quality ≥ 25) for about half of the historical remains fell within two standard deviations of the mean for the ancient samples (Figure 3A). Additionally, the average sizes of mapped human DNA fragments recovered from all of the historical remains were within the range for the ancient samples (Figure 3B). As the library preparation method used is known to impact the observed 3′ deamination [26], we compared the values on the 5′ end only. In this comparison, many of the historical remains had lower deamination rates than the average ancient sample (Figure 3C). In ancient DNA studies, a cut-off of 10% deamination is often used for the identification of authentic ancient DNA [38,43,47], and the majority of the historical samples exceeded this threshold. When looking at the rate of deamination along the DNA fragments, while the terminal base has the highest rate, elevated rates of deamination continue until the third terminal base for the historical samples (Figure 3D). This trend is known for ancient samples, but this observation indicates that it may be important to consider deamination during the downstream analyses of historical DNA samples, particularly when no DNA repair step is included. In addition, the similarity between the DNA recovered from the historical and ancient samples emphasizes the difficulty of performing identifications for forensic casework when remains may have been formalin treated or are poorly preserved.

### 3.3. Evaluation of Library Preparartion Protocols

The characterization of the DNA recovered from the historical remains confirmed the finding of previous studies that the DNA from these samples is highly degraded [4,15,48]. As the currently validated MPS workflow at AFMES-AFDIL uses a double-stranded library preparation method, it is likely that many of these DNA fragments are lost and not successfully converted into libraries. Therefore, success rates could be improved if a single-stranded library preparation method was used. We tested this hypothesis by taking three 5 µL aliquots from the ancient DNA extracts with a 56 °C incubation and preparing libraries with the KAPA Hyper Prep double-stranded kit (currently being validated for casework by AFMES-AFDIL) [15], the SRSLY single-stranded kit [49], and the single-stranded DNA library preparation method described in [18], which we refer to as the “MPI” method. For the double-stranded KAPA libraries, shotgun sequencing resulted in between 4 and 6997 unique nuclear DNA fragments recovered (two libraries failed during library prep). The two single-stranded DNA libraries resulted in between 23 and 143,388 unique DNA fragments for the SRSLY kit and 908 to 407,183 for the MPI method. It is important to note, however, that none of these libraries were sequenced to exhaustion. As with the extraction method comparison, we determined the relative informative sequences per sample for the three different library preparation methods based on the shotgun data (Figure 4). This confirmed that single-stranded library preparation methods were more effective at recovering informative sequences (*n* = 15 for the number of samples used for comparisons across the three different library preparation methods, Wilcoxon test with correction for multiple testing, both *p*-values < 2.0 × 10^−^^5^), although there was no significant difference between the two single-stranded methods (*p*-value = 0.29).

Next, we wanted to evaluate the mtDNA variant calling success rates of the two single-stranded methods for each sample to determine if there was a difference from previous casework attempts. In order to test this, each library was enriched for human mtDNA using hybridization capture. The resulting sequencing data was then processed both at the MPI and using a CLC workflow at the AFMES-AFDIL (see Section 2.5). No significant difference was observed in the percentage of sequences that mapped between the two library preparation methods (Figure 5A, Wilcox test with correction for multiple testing, *n* = 15, *p* = 0.57). However, during the processing of the SRSLY libraries, pig mtDNA was identified as mismapping to the human mtDNA genome (0 to 17 fragments, contributing up to 6.5% of sequences per sample library). Therefore, a “pig-out” alternative workflow was used to remove these sequences via competitive mapping for all the SRSLY libraries (Appendix A). The MPI method was found to result in more mtDNA sequences (Figure 5B, *p*-values adjusted for multiple testing 5.5 × 10^−^^4^ for MPI workflow and 7.6 × 10^−^^6^ for CLC workflow, *n* = 12 samples compared between library types, Wilcoxon test) with no significant difference in number of unique mtDNA sequences (*n* = 40 sample, library and workflow comparisons, *p*-value adjusted for multiple testing: 0.24, Wilcoxon test). Importantly, the maximum coverage for the negative controls was 0.3-fold for the CLC workflow, which is lower than typically observed in MPS data using the validated AFMES-AFDIL double-stranded DNA workflow [4], allowing for lowering thresholds below 10-fold for variant detection. For the CLC processing, two standard deviations above mean coverage of the controls was 0.38-fold. Thus the analytical threshold, above which sequence data can be distinguished from background noise, could be set as low as 2-fold (2×) average coverage. The SRSLY library preparation method resulted in 13 samples with an average coverage of at least 10-fold for the mtDNA genome and the MPI library preparation method resulted in 14 samples with at least 10-fold average coverage. While 10-fold was the previous coverage threshold, the low coverages observed in the negative controls supported decreasing this threshold comfortably to 5-fold, well above the 2-fold level of background noise. All samples resulted in at least 5-fold average coverage using either single-stranded library preparation technique (Figure 6, Appendix A).

For variant calling, only the CLC workflow was used as it applies forensic standards to haplotype reporting that have been previously validated for forensic casework use. When compared with previously generated casework results, the low-coverage and high-coverage sample data were nearly 100% concordant (Table 3 and Appendix A). It is important to note that the previous casework results were generated from two DNA libraries comprised of roughly 1.0 g of bone powder each; whereas the data in this study utilized DNA libraries comprised of 0.01 g of bone powder. The only discordance was observed in the high coverage sample 2255 at nucleotide position (np) 6755. Heteroplasmy (G6755R) was called in the 473-fold casework data due to a minor guanine (G) variant at 21.1% frequency. However, heteroplasmy was not observed above the set thresholds in either the MPI or the SRSLY data, which both produced a G6755A variant. It is noted that the MPI data exhibited 15-fold coverage at this position, with 13 (87%) of reads exhibiting adenine (A) at this position and two reads (13%) producing a guanine (G) base. The SRSLY data had 11× coverage at np 6755 and produced 100% adenine. This decreased observation of guanine at this position may be due to a combination of lower coverage and cytosine deamination. It is possible that higher coverage, gained through either higher DNA input into library or through combining data produced from multiple libraries and/or sequencing events, would result in improved heteroplasmy detection at this position in either of the single-stranded DNA libraries.

The seven samples that failed to produce reportable mtDNA profiles in routine casework (at a 10-fold coverage threshold) yielded concordant, partial profiles when comparing the 5-fold MPI and SRSLY data. Despite partial ranges, the predicted haplogroups (Table 3) were almost identical between the MPI and SRSLY data for all seven samples. The reason that they are not exactly identical is due to missed/uncalled variants in low coverage regions, in which the minimum 5-fold (5×) coverage threshold was met but the minimum variant count of 4 was not met—either due to cytosine deamination and/or low-level background noise. It is likely that, when merged with sequence data from a second/replicate library, the coverage and variant detection will improve. This merging of data from two independent libraries, as is routinely done in AFMES-AFDIL casework, would likely make these profiles reportable. Notably, only two of the seven failed samples (1837 and 1964) exceeded 5-fold average coverage in routine DNA casework. Thus at a minimum, the MPI and SRSLY methods improved the mtDNA profiling success of the bottom third (*n* = 5) of the poorest quality AFMES-AFDIL samples, from 0% to 100%. In other words, the previous method did not even produce partial mtDNA profiles at a 5-foldcoverage level that could be reported; whereas the ssDNA methods both yielded sufficient coverage for haplogrouping and distinguishing mtDNA profiles.

## 4. Discussion

The identification of historical remains usually falls within the scope of the forensic science community, particularly with remains from armed military conflicts. The DNA recovered from the samples in this set, which had varied preservation conditions, including formalin treatment, showed similar properties to DNA recovered from remains over 30,000 years old. While it has been previously demonstrated with both artificially degraded DNA and forensic casework samples that ancient DNA extraction methods can increase DNA yields in historical cases [28,30], this study clearly demonstrates the benefits of making this change for the disinterred remains of WWII and Korean War service members. The degraded nature of the DNA recovered from the remains investigated here, possibly facilitated by the powder application and/or formalin-treatment they were subjected to [25,26], made them clear candidates for single-stranded DNA library preparation. For the seven samples that had previously failed to result in data that met the required 10-fold average coverage for casework, six passed this threshold with at least one of the two single-stranded DNA library preparation methods. Importantly, these methods also resulted in cleaner negative controls, which allowed for lowering of thresholds for data reporting, a benefit when working with degraded samples that contain limited amounts of data. In this study, the lowering of the average coverage threshold to 5-fold enabled the reporting of two more samples, which would have failed if we had maintained the previous 10-fold threshold, allowing all samples to be reportable.

Typically, in forensic DNA analysis, analytical thresholds are established based on 2–3 standard deviations above the mean coverage in negative controls [4]. In this study that would place the average coverage threshold at 2-fold. However, we raised this threshold to 5-fold, in order to minimize the reporting of background noise, low-level variants, and contamination. This also raises an important discussion about the detection of contamination. Although both the ancient DNA and forensic fields follow similar clean room practices and maintain negative controls throughout processing, the methods for identification of contamination in sample data differs. Expected elevated rates of deamination in ancient samples allow for the examination of deamination patterns to determine the presence of endogenous DNA and estimate amounts of modern human contamination [41,42], which may result in decisions to restrict analyses to only fragments that have deamination. While the samples in this study showed elevated deamination rates, this is not the case for all forensic samples and cannot be relied upon for confirmation of endogenous DNA content. For this reason, the forensic field has established three pathways for detecting contamination. First, DNA profiles from all staff members and individuals known to have handled the remains are maintained allowing for source identification from any contamination that may arise from the handling of the remains during processing. Profiles among unidentified remains can also be compared this way in case cross-contamination occurs due to commingling of remains. Second, all results from degraded skeletal remains are replicated from a new/second bone powder aliquot that is taken through DNA extraction, downstream processing, and analysis. Finally, each profile is evaluated as being either single-source or a mixture using interpretation parameters established through comprehensive validation studies. For mtDNA, mixed/contaminated DNA samples are known to cause a reduction in variant frequency as well as an excess of heteroplasmy (>3 nucleotide positions across the mitogenome) [4]. Mixed DNA samples are not reportable, and the laboratory work will be repeated in an attempt to determine the source of the mixture. If the bone itself is contaminated, as evidenced by a repeat mixture after a second DNA extraction, the sample results will not be reported. Determining contamination within the bone requires having multi-fold coverage across the mitogenome, which is why we decided to use a threshold of 5-fold. Attempts to lower this threshold further would require the development of other bioinformatic tools for the identification of mixtures from low-coverage data.

Although exogenous human DNA is highly scrutinized, as described above, low-level contaminants from non-human sources can also impact the accuracy of variant calling. We were able to develop a pipeline to remove the identified pig mtDNA contaminant, but currently validated workflows at the AFMES-AFDIL do not include these steps or a way to easily screen for the presence of non-human DNA. There are different taxonomic classifiers and methods used in other fields to identify different species and families in sequence data [39,40,50]. Based on the pig DNA observed in the SRSLY data, it may be worthwhile considering integrating one of these into workflows when working with historical casework. Additionally, the presence of deamination in all samples raises the question of how to ensure this does not impact downstream variant calling. In the presented analysis, variants that appeared to be due to deaminated bases were removed from the profile for concordance purposes. For the typical casework workflow, the impacts of deamination would likely be mitigated by merging the data from multiple extractions and/or libraries. In ancient DNA studies, different methods are used to minimize the impact of deamination on downstream analyses. For example, the terminal three positions of all fragments are often masked or, in extreme cases, only variants that are transversions are used [51,52]. A previous sensitivity study on the impact of DNA degradation on STR and SNP concordance with Verogen’s ForenSeq kit also found differences in concordance and heterozygosity balance in comparison to studies with non-degraded DNA [28], indicating that considerations for the analysis of degraded DNA across platforms is of importance. Future studies on the impacts of low-input and degraded DNA for downstream forensics analyses will both help to increase the accuracy when working with more challenging data and open up the possibility of producing accurate, identification informative data from previously inaccessible samples.

In conclusion, we have identified that an altered ancient DNA extraction protocol coupled with single-stranded library preparation increases the amount of recovered mtDNA from disinterred military service member skeletal remains. These findings emphasize the value of exchanging methods between the ancient DNA and forensic fields [29], which are beneficial in both directions as shown from the ancient DNA community integrating bleach treatment to remove contamination from samples [53,54]. The integration of the methods in this study into casework samples with highly degraded DNA has the potential to increase success rates for identifying historical remains.

## Figures and Tables

**Figure 1 genes-13-00129-f001:**
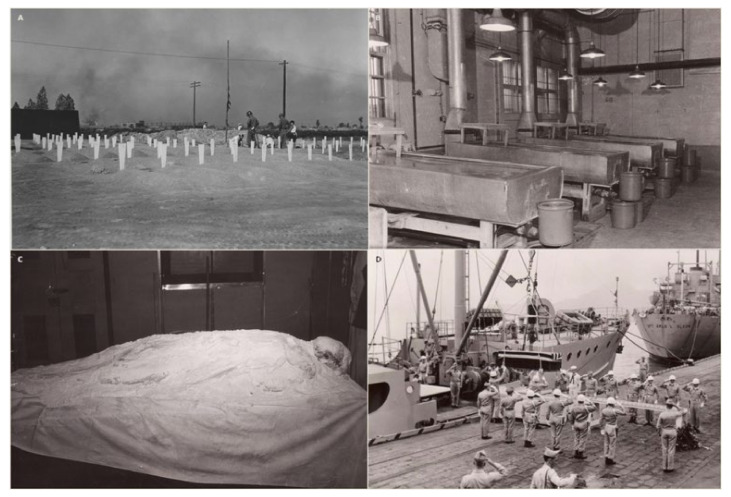
These historical photographs show some of the important events that occurred after service members were killed in action during the Korean War. (**A**) In this photograph, taken on 19 August 1950, a battlefield cemetery in Daegu (formerly Taegu), Korea, contains dozens of temporary graves where the bodies of soldiers killed in action were laid to rest. This image was reprinted with permission of the U.S. Army Quartermaster Museum in Fort Lee, Virginia. (**B**) This photograph of the Kokura embalming laboratory was originally printed in the American Graves Registration Service Group 8204th Army Unit APO 3 Brochure, April 1955. Examination tables were equipped with exhaust systems for formaldehyde treatment. (**C**) These remains from a Korean War service member were covered with a powdered hardening compound prior to shipment to the United States. This photograph was reprinted courtesy of the DPAA. (**D**) In Operation GLORY, the first of the United Nations war dead from North Korea were received at Moji Port near Kokura, Japan (ca. fall of 1954). This photograph was originally printed in the American Graves Registration Service Group 8204th Army Unit APO 3 Brochure, April 1955.

**Figure 2 genes-13-00129-f002:**
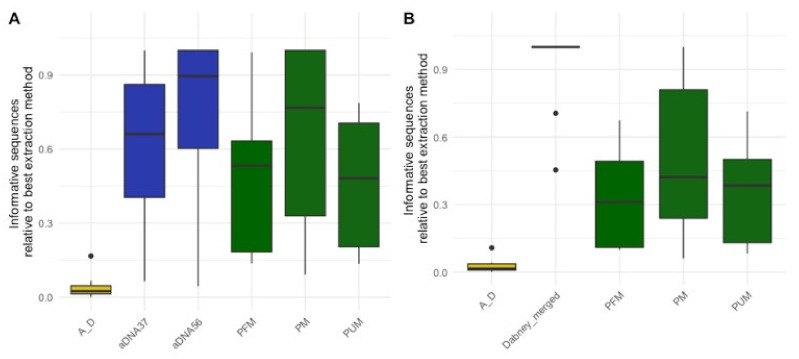
Relative informative sequences recovered per extraction protocol across 15 historical skeletal remains with the ancient DNA extraction fractions (**A**) separate and (**B**) merged. The total informative sequences per library was calculated by multiplying the number of DNA molecules in each library as determined by qPCR by the proportion of sequences longer than 35 bp that mapped to the human reference genome. The relative informative sequences were then calculated by taking the ratio of total informative sequences for each extraction/sample combination to the total number of informative sequences from the best performing extraction protocol per sample. The box plots follow the standard Tukey representation to show the distribution of the relative informative sequences per extraction type (*n* = 15). The significance of different relative amounts recovered per extraction was evaluated with a Wilcoxon test with a correction for multiple testing. The ancient DNA protocols (aDNA37: 37 °C digestion; aDNA56: 56 °C digestion; Dabney merged: merging of total informative sequences per extraction/sample from aDNA37 and aDNA56) are colored in blue; forensic protocols (PM: PCIA with Min Elute; PUM: PCIA with Min Elute and USER treatment; PFM: PCIA with Min Elute and FFPE treatment) in green; and the combined forensic ancient DNA protocol (A_D: AFDIL digestion with Dabney purification) in yellow.

**Figure 3 genes-13-00129-f003:**
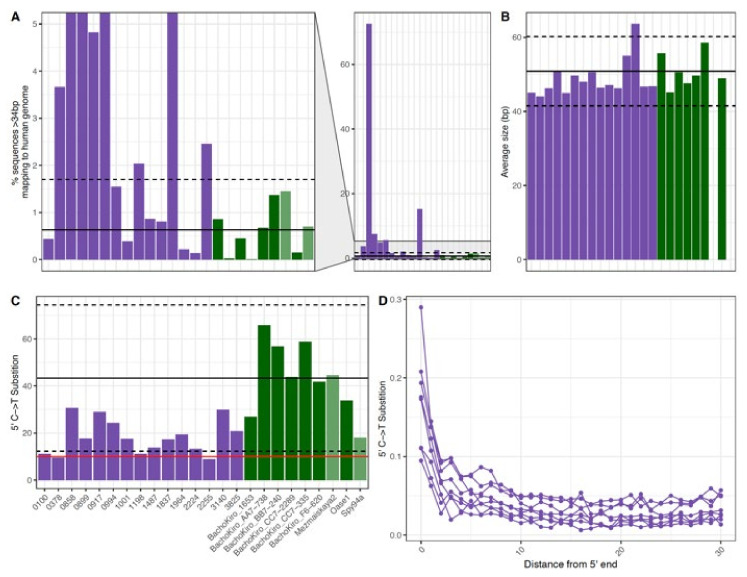
The (**A**) percent mapped (**B**) average fragment size (bp) and (**C**,**D**) observed 5′ C to T substitution frequencies of DNA recovered from historical and ancient DNA samples. Each bar represents a single sample for fifteen historical samples (purple) and seven Late Pleistocene modern human skeletal samples from ~37,000–46,000 years ago (dark green). In (**A**–**C**) the *x*-axis is the same for each and the black line represents the average for the ancient skeletal (dark green) samples and the dashed lines show two standard deviations above and below this average. In (**C**) the red line represents the 10% threshold typically used to determine if a sample contains ancient DNA. In (**D**) each point represents the 5′ C to T substitution frequency at a position from the 5′ end of DNA fragments recovered from a single sample. Note that only historical samples are shown in (**D**).

**Figure 4 genes-13-00129-f004:**
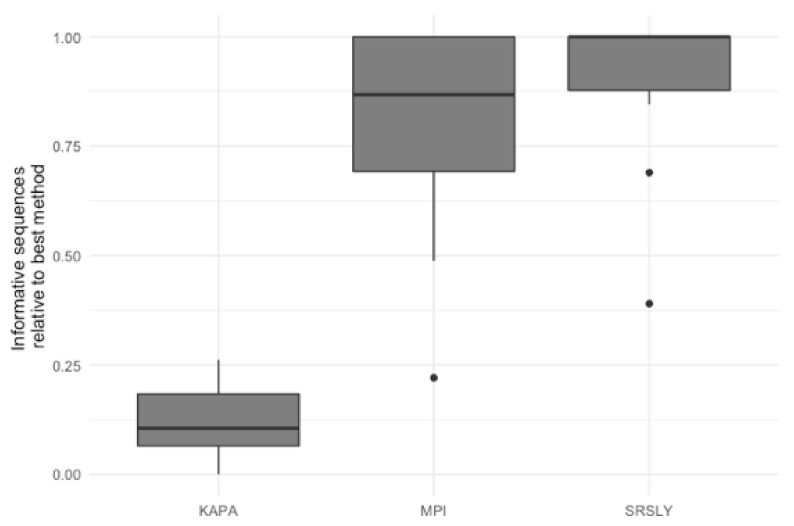
Relative informative nuclear DNA sequences recovered per library preparation protocol across 15 historical skeletal remains. The relative informative sequences per library was calculated as described in Appendix A. The box plots follow the standard Tukey representation to show the distribution of the relative informative sequences per extraction type (*n* = 15). The significance of different relative amounts recovered per library prep type was evaluated with a Wilcoxon test with a correction for multiple testing.

**Figure 5 genes-13-00129-f005:**
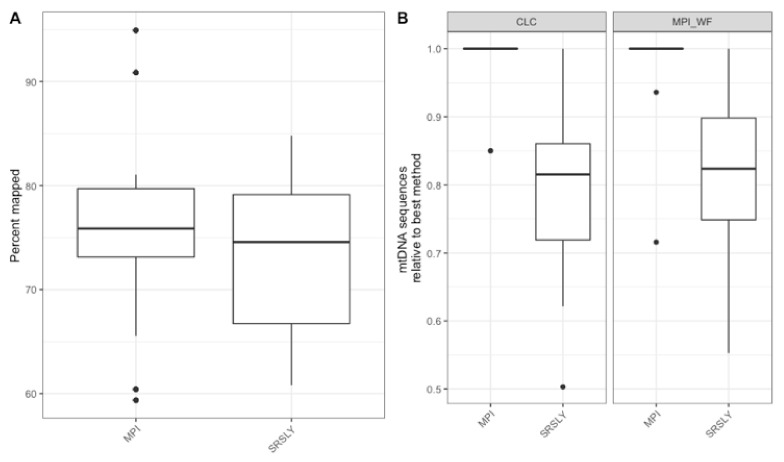
(**A**) The distribution of the percentage of sequences at least 30 bp long that mapped to the human mtDNA reference genome after capture using the MPI workflow. (**B**) The relative number of mapped mtDNA sequences across library preparation methods for both the CLC and MPI workflows. The box plots follow the standard Tukey representation to show the distribution of the relative informative sequences per extraction type (*n* = 15).

**Figure 6 genes-13-00129-f006:**
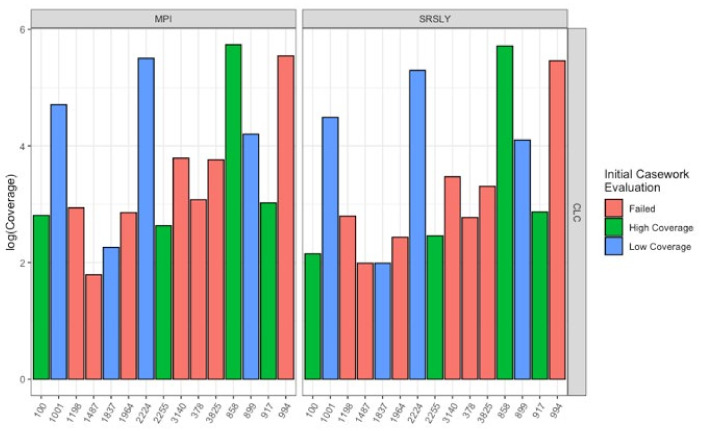
Average coverage of different qualities of samples with MPI and SRSLY library preparations after enrichment for human mtDNA using the CLC workflow. Samples are colored based on the quality as determined from the initial casework evaluation before changing the extraction and library preparation methods (Table 1). The left plots underwent MPI library preparation and the right plots SRSLY library preparation.

**Table 1 genes-13-00129-t001:** Sample descriptions and case contexts. World War II (WWII) conflict locations are shown, as well as the cemetery of disinterment. Formalin treatment was not documented, but was noted as being likely in certain cases. Powder application was evidenced by visible powder on the remains. Casework results were based on the coverage of the revised Cambridge Reference Sequence (rCRS) [31] produced from the same skeletal sample using the methods outlined in [4].

Sample ID	Skeletal Element	Context	Cemetery of Disinterment	Formalin Treatment	Powder Application	Casework Results
2224	Tibia	WWII-Tarawa	NMCP ^1^	Unknown	Not visible	High Coverage
0858	Temporal	WWII-USS Oklahoma	NMCP ^1^	No, Oil Soaked	Likely	High Coverage
0994	Femur	WWII-Buna	MAC ^2^	Unknown	Not visible	High Coverage
2255	Humerus	WWII-Cabanatuan	MAC ^2^	Unknown	Not visible	High Coverage
0899	Humerus	WWII-Tarawa	NMCP ^1^	Unknown	Yes	Low Coverage
0100	Femur	Korea	NMCP ^1^	Likely	Yes	Low Coverage
1001	Os coxa	WWII-Yugoslavia	SRAC ^3^	Unknown	Not visible	Low Coverage
3140	Radius	WWII-Burma	NMCP ^1^	Unknown	Yes	Low Coverage
0378	Femur	WWII-Tarawa	NMCP ^1^	Unknown	Yes	Failed
1837	Tibia	WWII-Tarawa	NMCP ^1^	Unknown	Yes	Failed
1198	Tibia	Korea	NMCP ^1^	Likely	Yes	Failed
1964	Humerus	Korea	NMCP ^1^	Likely	Yes	Failed
0917	Tibia	Korea	NMCP ^1^	Likely	Yes	Failed
3825	Radius	Korea	NMCP ^1^	Likely	Yes	Failed
1487	Femur	WWII-Italy	SRAC ^3^	Unknown	Yes	Failed

^1^ National Memorial Cemetery of the Pacific, Honolulu, Hawaii, USA; ^2^ Manila American Cemetery, Philippines; ^3^ Sicily-Rome American Cemetery, Italy.

**Table 2 genes-13-00129-t002:** DNA extraction methods tested.

Method	Bone Powder (g)	Digestion Buffer	Digestion Buffer Volume (mL)	Proteinase K (20 mg/mL) (µL)	Incubation Temperature (°C)	DNA Purification	Repair Protocol
AFDIL (PM)	1.0	Demin buffer ^1^	7.5	200	56	PCIA with buffer exchange	NA (MinElute purification)
AFDIL-USER (PUM)	1.0	Demin buffer ^1^	7.5	200	56	PCIA with buffer exchange	USER (NEB, Ipswich, MA, USA)
AFDIL-FFPE (PFM)	1.0	Demin buffer ^1^	7.5	200	56	PCIA with buffer exchange	NEBNext FFPE DNA Repair Mix (NEB)
Dabney—37 (aDNA37)	0.2	Dabney buffer ^2^	1.0	25	37	Silica column and PB Buffer	NA
Dabney—56 (aDNA56)	Dabney 37 remaining pellet	Dabney buffer ^2^	1.0	25	56	Silica column and PB Buffer	NA
AFDIL-Dabney (A_D)	0.2	Demin buffer ^1^	4.0	200	56	Silica column and PB Buffer	NA

^1^ Demin buffer: 0.5 M EDTA, 1% lauroyl sarcosine; ^2^ Dabney buffer: 0.45 M, 0.5% Tween 20.

**Table 3 genes-13-00129-t003:** Haplotype comparison between previously generated forensic casework data using the methods described in [4] and two single-stranded DNA library preparations (the Max Planck Institute (MPI) [18] and SRSLY [49]) after enrichment for human mtDNA. The AFMES-AFDIL casework minimum coverage threshold was 10-fold and the maximum number of reportable bases was 16,507, as regions of length heteroplasmy are not reported in routine casework. The MPI and SRSLY coverage thresholds were 5-fold. The SRSLY bioinformatics workflow required removal of contaminating *Sus scrofa* mitochondrial DNA (mtDNA) sequences prior to mapping to the rCRS [31].

Sample ID	Casework		MPI			SRSLY (Pig Out)		
	Reported Bases	Predicted Haplogroup	Reported Bases	Predicted Haplogroup	Discordant Sites (Compared to Casework)	Reported Bases	Predicted Haplogroup	Discordant Sites (Compared to Casework)
High Coverage								
2224	16507	H18	16569	H18	0	16568	H18	0
0858	16507	V3c	16568	V3c	0	16569	V3c	0
0994	16507	J1c4b	16569	J1c4b	0	16569	J1c4b	0
2255	16507	J1c3	16268	J1c3/J1c3h/J1c3k	1	16144	J1c3	1
Low Coverage								
0899	16507	H4a1a4b	16565	H4a1a4b	0	16563	H4a1a4b	0
0100	16183	H24a	16181	H24a	0	13585	H24/H24a	0
1001	15765	H1c6	16566	H1c6	0	16567	H1c6	0
3140	16505	U5b1b1-16192	16555	U5b1b1- 16192	0	16515	U5b1b1-16192	0
Failed in Casework								
0378	NA	NA	15951	U5a1a1+16362/U5a1a1d/U5a1a1d1	NA	15681	U5a1a1+16362/U5a1a1d/U5a1a1d1	NA
1837	NA	NA	13504	K1c2	NA	11526	K1c2	NA
1198	NA	NA	16091	H1c3b	NA	15821	H1c3b	NA
1964	NA	NA	16010	L2a1c+16086/L2a1c3b	NA	14871	L2a1c+16086	NA
0917	NA	NA	16386	K1a1b2b	NA	16004	K1a1b2b	NA
3825	NA	NA	16550	H11a	NA	16319	H11a	NA
1487	NA	NA	10715	K1a (and subhaplogroups of K1a)	NA	12009	K1a9	NA

## Data Availability

Data are stored at the AFMES-AFDIL and may be made available to approved laboratories upon written request.

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
