# Peer review of "Ancient DNA Methods Improve Forensic DNA Profiling of Korean War and World War II Unknowns"

_genes, 2022, doi:10.3390/genes13010129_

Round 1

Reviewer 1 Report

This manuscript validated several methods in ancient and forensic DNA to identify mitogenomes of skeletons of war dead in WWII. Although ancient and forensic DNA studies target degraded DNA, DNA extraction and sequencing (or genotyping) methods differ between the two research fields. Methods in Ancient DNA have more advantages for more degraded (older) samples and the use of massively parallel sequencing (MPS, or just say NGS) than those in forensic DNA. The authors of the manuscript focus on this gap; not so many studies compare the performance of those methods using the same materials. In the analysis, the authors showed standard methods in ancient DNA (DNA extraction by Dabney's method and single-stranded DNA library preparation) are superior to those in conventional forensic DNA in terms of the required weight of materials and informative sequences.  This study would contribute to multiple research fields, particularly the introduction of MPS to forensic or historical sample studies.  However, I found some minor questions about their data.

1. line 304 on page 8. The authors describe, "When processing the SRSLY libraries, contaminant pig DNA was identified as mismapping to the human mtDNA reference genome (rCRS). To remove these sequences a “pig-out” pipeline was added to both the CLC and MPI workflows as described in SI Text S1."  
I am very confused because the description of pig DNA suddenly appears in this sentence.  The authors suspect that some pig DNA contaminated into SRSLY library. In SI, the authors compared the mapping quality of the reads to human mtDNA before/after removal of potentially pig DNA fragments. I respect the authors to find this; however, the authors did not show direct evidence of the suspected pig sequences. Please show more solid evidence of pig DNA sequences, for example, alignments of suspected reads against pig and human mtDNA, or by any appropriate manner. 

2. Figures 3A-B lack labels of the x-axis. I guess these labels are the same as C, but the scales (length) of the three barplots are different. It is not reader-friendly. Please modify the labels. 

3.  In Figure 3D, only historical samples are shown, while both historical and Late Pleistocene samples are shown in Figure 3A-C.  Why does Figure 3D lack the older samples? 

4. Figure 5 is not clear what color indicates. What is "the previous testing"? For example, red indicates "failed" samples, but log(Coverage) of sample 0994 is the second-highest in the SRSLY pannel. Similarly, Sample #2224 is blue (suggesting "low" coverage), while the log(Coverage) of 2224 is the third highest in SRSLY. These two samples look good quality. 

Author Response

  1. line 304 on page 8. The authors describe, "When processing the SRSLY libraries, contaminant pig DNA was identified as mismapping to the human mtDNA reference genome (rCRS). To remove these sequences a “pig-out” pipeline was added to both the CLC and MPI workflows as described in SI Text S1."  
    I am very confused because the description of pig DNA suddenly appears in this sentence.  The authors suspect that some pig DNA contaminated into SRSLY library. In SI, the authors compared the mapping quality of the reads to human mtDNA before/after removal of potentially pig DNA fragments. I respect the authors to find this; however, the authors did not show direct evidence of the suspected pig sequences. Please show more solid evidence of pig DNA sequences, for example, alignments of suspected reads against pig and human mtDNA, or by any appropriate manner.
    1.This is a key point to understand so we appreciate the reviewer’s attention to it. In order to clarify that we used taxonomic classifiers to confirm the presence of pig DNA, and that the sequence alignments were visually inspected, we expanded this section as follows:

When processing the SRSLY libraries, contaminant pig DNA was identified as mismapping to the human mtDNA reference genome (rCRS), resulting in spikes of coverage and unexpected variants (Supplementary Figures 1 and 2). These spikes were not observed in the MPI libraries and therefore we assumed the source was likely an artefact of the library preparation and not the samples themselves. The presence of pig sequences in these libraries was confirmed using a pipeline that uses BLAST [38] and MEGAN [39] to assign sequences from sediment samples to different mammalian families (Supplementary Figure 3). To remove these sequences a “pig-out” pipeline was added to both the CLC and MPI workflows as described in SI Text S1.  

2. Figures 3A-B lack labels of the x-axis. I guess these labels are the same as C, but the scales (length) of the three barplots are different. It is not reader-friendly. Please modify the labels. 

Yes, the reviewer is correct that the labels for 3A-B are the same as C. We decided to not repeat the labels in order to save space and limit repeated information. We have updated the figure legend to clarify this. We also noticed that the mean and standard deviation lines for 3B were missing and so we updated the figure.

  1. In Figure 3D, only historical samples are shown, while both historical and Late Pleistocene samples are shown in Figure 3A-C.  Why does Figure 3D lack the older samples? 

We utilized published data for figure 3(A-D). Only the terminal position deamination data from these Late Pleistocene samples was published. Deamination data across each fragment was not published, so they were not included in 3D. This is now noted in the figure legend.

The reason for choosing these particular Late Pleistocene samples, as opposed to others with published deamination data, is that these were most comparable to the historical samples in terms of downstream methods, including DNA extraction, library preparation, and capture procedures.

4. Figure 5 is not clear what color indicates. What is "the previous testing"? For example, red indicates "failed" samples, but log(Coverage) of sample 0994 is the second-highest in the SRSLY pannel. Similarly, Sample #2224 is blue (suggesting "low" coverage), while the log(Coverage) of 2224 is the third highest in SRSLY. These two samples look good quality. 

Thank you for pointing out that this was not clear. By previous testing we meant the results when using the original workflow methods and not the results when using the ancient DNA extraction protocol with single stranded DNA library preparation. In order to clarify this we have added a legend label and edited the figure legend.

Reviewer 2 Report

In this contribution Zavala and colleagues test the performance of ancient DNA extraction and library preparation methods for recovering mitochondrial DNA (mtDNA) sequences from highly degraded forensic samples. Specifically, the authors compare and contrast the performance of three extraction and library preparation protocols (including single versus double stranded methods) for recovering mtDNA SNP profiles from the skeletal remains of World War II and Korean War service members. It is important to note that many of the samples tested had previously failed to generate useable information due to extensive post-mortem degradation and funerary chemical treatments. The authors find that applying ancient DNA protocols designed for recovering DNA fragments less than 50bp in length, plus single stranded library preparation, enrichment, and next-generation sequencing approaches (AKA massively parallel sequencing) yields much better results than using standard forensic approaches which rely on PCR and Sanger sequencing or double stranded library preparation. They demonstrate that a major reason why these other methods fail is because these highly degraded forensic samples have all the traits of much more ancient samples (e.g., small fragment sizes, deamination, cross-linking). Thus, ancient DNA methods, which are much better equipped to handle these challenges are much more appropriate for such forensic samples and therefore may lead to more positive identifications.

I find this article timely and pertinent to the audience of this journal. The topic will be of interest to researchers interested in anthropological genetics, forensic human identification, paleogenomics, and potentially, war history. I have no issues with the methods used in this study. The experimental design is sound, and the researchers provide sufficient detail for independent replication of their findings. I especially appreciated the authors’ nuanced consideration of the challenges of implementing these methods in standard forensic casework in the Discussion. For all these reasons I think the article will be an important addition to the peer reviewed literature that may eventually aid forensic laboratories and agencies in adopting and validating the use of ancient DNA methods in forensic caseworks. The figures are appropriate, and the authors have covered most of the current literature on this subject in their bibliography. Overall, this is a very good article and I enjoyed reading it very much! Below, I offer just a few suggestions to aid the authors in strengthening the manuscript.

Authors, I hope that these comments are helpful in ensuring successful publication. Thank you for the opportunity to review your research. Good luck with the revision process!

Introduction:

  • Can the authors add some more information on the climactic conditions that prevail in the areas where the tested remains were buried (e.g., Korea and Hawaii) and how they think these climactic conditions may have impacted (accelerated?) the degradation process?
  • Given that shotgun data was generated, the authors could also perform some calculations as in Allentoft et al. (2012) to determine the average half-life of DNA and rate of decay in the historic samples. I suspect the results will show that these historic remains have similar degradation patterns to much more ancient samples. (https://royalsocietypublishing.org/doi/10.1098/rspb.2012.1745)

Materials and Methods:

  • I would have liked to see some comparisons between the different skeletal elements examined in this study. Do the authors see any differences in DNA degradation across samples collected from the tibia, temporal bone, femur or humerus for instance?
  • Additionally, I was wondering why teeth were not considered for testing, given that the enamel is likely to preserve the DNA.
  • Were the DNA extracts quantified before library preparation? If so, I was wondering if DNA amounts detected were below the threshold of fluorometer quantification or if they were, indeed, detectable.

Results:

  • Typo in line 453: “calliing” should be “calling”
  • This may be a formatting issue upon submission and outside of the control of the authors but, could Table 3 be all in one page instead of split into two pages?

Supplemental Information:

  • Typo on page 3, first line: “difference” should be “different”

Author Response

Introduction:

  • Can the authors add some more information on the climactic conditions that prevail in the areas where the tested remains were buried (e.g., Korea and Hawaii) and how they think these climactic conditions may have impacted (accelerated?) the degradation process?

Thank you for bringing this up, this is important contextual information. We have added the following sentence:

In addition, Hawaii has a warm and wet environment, while Korea is climatically diverse with temperatures fluctuating from freezing to extremely hot. Such environmental conditions involving high heat have been found to increase DNA degradation, specifically deamination and fragmentation rates[15].

  • Given that shotgun data was generated, the authors could also perform some calculations as in Allentoft et al. (2012) to determine the average half-life of DNA and rate of decay in the historic samples. I suspect the results will show that these historic remains have similar degradation patterns to much more ancient samples. (https://royalsocietypublishing.org/doi/10.1098/rspb.2012.1745)

This is a good idea and an area we would like to explore in the future. Unfortunately, these samples all come from around the same time (~70 to 80 years ago), while the samples from the Allentoft paper cover around 7,000 years. There is a knowledge gap on decay rates for younger samples (less than 1,000 years old), but we would need more samples across this time period to explore this further.

Materials and Methods:

  • I would have liked to see some comparisons between the different skeletal elements examined in this study. Do the authors see any differences in DNA degradation across samples collected from the tibia, temporal bone, femur or humerus for instance?

This is an interesting and important question that we had initially considered. However, as each of the elements come from different individuals, as well as different contexts with different preservation conditions, we thought that these factors may confound any correlations among the different skeletal elements. Comparisons of elements within single skeletons across multiple skeletons would provide further insight into this question, which is an area that could be explored further in future studies.

  • Additionally, I was wondering why teeth were not considered for testing, given that the enamel is likely to preserve the DNA.

For this particular study, it was important to have samples that ranged in DNA quality. Therefore, the samples were selected based on their known performance from previous casework processing. As a result, only samples with remaining tissue/powder after casework processing could be utilized. As tooth powder is often consumed during casework processing, there is usually nothing left for research.

However, due to the interest in teeth from formalin preserved samples, we are now conducting a separate study to examine DNA recovery from their cementum/dentine. This separate study involves samples that were not previously tested in routine casework, with ample tissue for sampling.

Were the DNA extracts quantified before library preparation? If so, I was wondering if DNA amounts detected were below the threshold of fluorometer quantification or if they were, indeed, detectable.

We did quantify the extracts using a Qubit HS fluorometric assay, but this represents mostly microbial DNA. In typical caseworking processing this amount is used to prevent overloading the library
preparation assay with too much DNA (as described in Marshall et al., 2017). However, as we did not use this information for calculating library input in these experiments and the qPCR of the libraries is
more reflective of usable DNA content, we decided not to include this information.

Results:

  • Typo in line 453: “calliing” should be “calling”

Thank you – this is fixed.

  • This may be a formatting issue upon submission and outside of the control of the authors but, could Table 3 be all in one page instead of split into two pages?

Thank you for pointing this out, it is currently on one page and we will check that it stays this way.

Supplemental Information:

  • Typo on page 3, first line: “difference” should be “different”

Thank you – this is fixed.